# Bounding and Approximating Intersectional Fairness through Marginal Fairness

**Mathieu Molina**
Inria
FairPlay team
91120 Palaiseau, France
`mathieu.molina@inria.fr`

**Patrick Loiseau**
Inria
FairPlay team
91120 Palaiseau, France
`patrick.loiseau@inria.fr`

## Abstract

Discrimination in machine learning often arises along multiple dimensions (a.k.a. protected attributes); it is then desirable to ensure *intersectional fairness*—i.e., that no subgroup is discriminated against. It is known that ensuring *marginal fairness* for every dimension independently is not sufficient in general. Due to the exponential number of subgroups, however, directly measuring intersectional fairness from data is impossible. In this paper, our primary goal is to understand in detail the relationship between marginal and intersectional fairness through statistical analysis. We first identify a set of sufficient conditions under which an exact relationship can be obtained. Then, we prove bounds (easily computable through marginal fairness and other meaningful statistical quantities) in high-probability on intersectional fairness in the general case. Beyond their descriptive value, we show that these theoretical bounds can be leveraged to derive a heuristic improving the approximation and bounds of intersectional fairness by choosing, in a relevant manner, protected attributes for which we describe intersectional subgroups. Finally, we test the performance of our approximations and bounds on real and synthetic data-sets.

## 1 Introduction

Research on fairness in machine learning has been very active in recent years, in particular on fair classification under *group fairness* notions, see e.g., [16, 30, 34, 33, 7, 28]. Such notions define demographic groups based on so-called *protected attributes* (e.g., gender, race, religion), and impose that some statistical quantity be constant across the groups. For instance, demographic parity imposes that the class-1 classification rate is the same for all groups, but other notions were defined such as equal opportunity [16] or calibration by group [7]—see a survey in [4]. As exact fairness is too constraining, one often measures *unfairness*, which roughly quantifies the distance to the fairness constraint.

Most works on fair classification consider a single protected attribute and hence only two (or a small number of) groups. Then, they use measures of unfairness to evaluate and penalize classifiers in order make them more fair. This is making an implicit but very fundamental assumption that one can estimate the unfairness measure from the data at hand. With only a few groups, this assumption is indeed easily satisfied as there are sufficiently many data points for each group.

In many—if not most—real-world applications, there are multiple protected attributes (typically 10-20) along which discrimination is prohibited [1, 2]. It is then desirable to consider the strong notion of *intersectional fairness*, which roughly specifies that no subgroup (defined by an arbitrary combination of protected attributes) is unfavorably treated. In that case, however, estimating the unfairness measure becomes very challenging: as the number of groups is exponentially large (e.g., $2^{10}$ for 10 binary protected attributes), it is very likely that the dataset has at least one subgroup

36th Conference on Neural Information Processing Systems (NeurIPS 2022).

for which there are very few (or zero) data point. A potential solution is to treat each protected attribute separately through its *marginal unfairness* (which is easy to estimate); but it was observed in several real-world and algorithmic examples that it is not sufficient to ensure intersectional fairness [8, 5, 21, 22]. This raises the question: *How to estimate intersectional fairness from data, and what is its precise relation to marginal fairness?* To date, only very few papers have tackled this issue. [21, 22] adopt a definition of intersectional fairness that weights the unfairness of each group by its size. This allows them to get large-samples generalization guarantees of empirical estimates (hence solving the estimation issue), but then it does not protect minorities since it allows a very high unfairness for tiny subgroups—which is contradictory to the intuitively desired behavior.

[17] makes a similar assumption by considering only subgroups above a minimum size, which eases estimate generalization. [14] on the other hand uses the more natural definition of intersectional fairness based on the worst treated group irrespective of its size; but they consider only a few protected attributes, precisely to have enough data points on each subgroup to estimate intersectional unfairness. [13] extends this work by proposing methods to interpolate for subgroups for which too few points are available, based on Bayesian machine learning models. However, this work is empirical and does not give any guarantee on the estimates obtained. In this paper, we also use the natural (strong) definition of intersectional fairness but we take instead a purely statistical approach. We view the protected attributes as random variables to understand intersectional fairness and how it related to marginal fairness more finely.

**Contributions:** We identify sufficient conditions under which intersectional fairness can be exactly derived from marginal densities, which clarifies when marginal unfairness is a good estimate of intersectional unfairness. We prove probabilistic bounds on intersectional unfairness based on marginal densities and independence measures of the protected attributes, that we show are easy to estimate. We propose a method to improve the approximation of intersectional unfairness and the theoretical bound based on grouping carefully some of the protected attributes together, which we do through a heuristic by leveraging the independence measures exhibited in our bounds. We perform experiments on real and synthetic datasets that illustrate the performance of our approach. In particular, we show that grouping with our heuristic does improve the approximation of intersectional unfairness. To the best of our knowledge, our work is the first work to exploit statistical information to better understand and estimate intersectional (un)fairness. Our work is fairly general and can be instantiated for a variety of standard fairness notions (demographic parity, equal opportunity, etc.). For simplicity, we focus on discrete protected attributes and on classification, but most of the core results can be extended to other cases.

**Further Related Works:** [31] proposes a unified framework to train a fair classifier under intersectional fairness metrics, but without taking into account regimes with sparse group membership data. Some works propose to audit the accuracy of fairness metrics in contexts other than intersectionality, when there are missing data [35] or when there are unlabeled examples [18]. Others tackle the problem of intersectionality beyond group fairness, e.g., [32] considers causal intersectional fairness. Finally there has been some interest [24, 10] in a different formulation of intersectional group fairness as a multi-objective optimization problem where each objective is the discrimination faced by a given protected group. Another interesting approach to fairness is individual fairness developed in [12], however this is quite different from group fairness metrics on which we focus on and our techniques do not apply.

## 2 Setting and Models

### 2.1 Basic Setting

*Notational convention*: Wherever useful, for any two random variables $V$ and $W$, we will use the shorthand $p_V(v) = \Pr(V = v)$, $p_{V,W}(v, w) = \Pr(V = v, W = w)$ and $p_{V|W}(v \mid w) = \Pr(V = v \mid W = w)$.

Consider a multi-class classification task. A given individual is described by a tuple of random variables $(X, A, Y)$ drawn according to a distribution $\mathcal{D}$ where $X$ is the features vector, $Y$ is the label with values in $\mathcal{Y}$, and $A = (A_1, ..., A_d)$ is a $d$-tuple of protected attributes. The only variable used to make a prediction is $X$ and the only variable to measure unfairness is $A$, but otherwise there are no constraints and $A$ can be a part of $X$. We denote the support of $X, Y, A$ and $A_k$ for $1 \leq k \leq d$, by $\mathcal{X}, \mathcal{Y}, \mathcal{A}$ and $\mathcal{A}_k$ respectively. We assume that $\mathcal{A}$ is finite (hence discrete). For a deterministic classifier $h$, $\hat{Y} = h(X)$ is the predicted class for a random individual. The classifier $h$ is fixed, as we are interested in measuring its fairness and not finding a fair classifier.

To compare the discrimination between groups, we consider a second random variable $A'$ such that $(A' \mid \hat{Y})$ is independent and identically distributed (i.i.d.) to $(A \mid \hat{Y})$. Some authors look at the difference in the treatment of protected groups as a ratio [14], and some others as a difference [21]. Here we choose to study discrimination in terms of ratio. We further apply a logarithm to symmetrize the discrimination measure between two protected groups and for ease of computation. We will consider Statistical Parity for simplicity of exposition, but other group fairness metrics can be either derived directly or adapted using the methods developed in this paper (see Appendix A.1). We define our measure of unfairness as follows:

**Definition 2.1.** For a distribution $\mathcal{D}$ and a classifier $h$, we define the *intersectional unfairness* and the $k^{th}$ *protected attribute marginal unfairness* as:

$$u^* = \sup_{(y,a,a') \in \mathcal{Y} \times \mathcal{A}^2} u(y,a,a'), \quad \text{and} \quad u_k^* = \sup_{(y,a_k,a_k') \in \mathcal{Y} \times \mathcal{A}_k^2} u_k(y,a_k,a_k') \tag{1}$$

$$\text{with } u(y,a,a') = \left| \log\left( \frac{\Pr(\hat{Y}=y \mid A=a)}{\Pr(\hat{Y}=y \mid A'=a')} \right) \right|, \; u_k(y,a_k,a_k') = \left| \log\left( \frac{\Pr(\hat{Y}=y \mid A_k=a_k)}{\Pr(\hat{Y}=y \mid A_k'=a_k')} \right) \right|. \tag{2}$$

One could think that if the marginal unfairness of each protected attribute is smaller than some $\epsilon > 0$, then the overall unfairness is smaller than $\epsilon$; measuring $u^M = \sup_k u_k^*$ corresponds to this idea. As stated in the introduction this is not sufficient to describe unfairness and we can still have $u^* > u^M$. We can rewrite (1) as $u^* = \sup_{\mathcal{Y}} \log(\sup_{\mathcal{A}} p_{\hat{Y}|A} / \inf_{\mathcal{A}} p_{\hat{Y}|A})$, and similarly for $u_k^*$. This means that to measure unfairness we only need to analyze the function $p_{\hat{Y}|A}$.

## 2.2 Estimation of Unfairness

If we want to estimate unfairness, the most straightforward approach is to estimate the probability mass function $p_{A,\hat{Y}}$ and then to compute the unfairness over these estimated distributions. The main difficulty in estimating the unfairness is estimating $\inf p_{\hat{Y}|A}$, as we can upper bound the $\sup$ by 1, but we cannot easily lower bound the $\inf$. For a data-set of $n$ samples and $d$ protected attributes, we denote for $(a,y) \in \mathcal{A} \times \mathcal{Y}$ the counts by group and prediction as $N_{a,y} = \sum_{i=1}^n \mathbb{1}[(A^{(i)}, \hat{Y}^{(i)}) = (a,y)]$ where $(A^{(i)}, \hat{Y}^{(i)})$ is the $i$-th i.i.d. realization of $(A, \hat{Y})$. The empirical probability is then defined as $\hat{\Pr}(\hat{Y}=y, A=a) = N_{a,y}/n$. [14] shows in Theorem VIII.3 that the error made by using empirical estimates is decreasing in $N_a$, which means that there needs to be sufficient data for each protected group to estimate $u^*$. When there are many protected groups the probability that at least one group receives no sample is high, and in this case there is at least one $a$ in $\mathcal{A}$ for which the empirical probability $\hat{\Pr}(\hat{Y}=y \mid A=a)$ is undefined, hence the $\inf$ and $\sup$ cannot be computed. [13] and [14] alleviate this issue of 0-counts by using a Dirichlet prior of uniform parameter $\alpha > 0$. This yield the Bayesian estimates $(N_{a,y} + \alpha)/(n + |\mathcal{A}||\mathcal{Y}|\alpha)$, that are then used to compute the estimator $u_B$. They also propose other methods to estimate $p_{\hat{Y}|A}$ which empirically performs better, but without guarantees; whereas $u_B$ has the nice property that $u_B$ is a consistent estimator of $u^*$. This is because of the consistency of the Bayesian probability estimates and by applying the Continuous Mapping Theorem for $\max$ and $\min$ which are continuous. Note that the empirical estimator (with $\alpha = 0$) is also consistent, but has infinite bias.

Nonetheless, $u_B$ has the drawback that for a low amount of samples and when the number of protected groups is high, it is almost determined deterministically by the parameter $\alpha$ and cannot be trusted. If $N_a = 0$ for a protected group $a$, the estimated distribution is uniform on $\hat{Y} \mid A=a$ and this group does not affect the computation of the $\sup$ and $\inf$. Hence if the most discriminated group is among the undiscovered one, we risk making an important error on the estimation. When $N_a$ increases, we gain more information on the distribution of $\hat{Y} \mid A$. However, when $N_a$ is still low for all groups, the estimated distribution of the $\inf$ of $\hat{Y} \mid A=a$ is almost entirely determined by the prior parameter $\alpha$.

## 2.3 Probabilistic Unfairness

When the number of protected subgroups grows arbitrarily large, it may be useless to try to guarantee fairness for every single one of them, regardless on how many people this truly affects. Should a decision maker sacrifice any potential predictive performance in order to guarantee fairness? It could be argued that an algorithm which discriminates 1 person among a 1000 can be described as fair to an extent. We may even be able to directly compensate the small amount of persons discriminated against if possible. Let us consider another example: if a company has clients on which it leverages machine learning predictions to make decisions, it would seem very limiting to guarantee fairness

for clients among specific protected groups for whom we will almost never deal with. Nevertheless, if the underlying clients distribution changes, our decision making process should also reflect this change in terms of fairness. This motivates looking at unfairness probabilistically in $(\hat{Y}, A, A')$. To do that we define the random unfairness $U = u(\hat{Y}, A, A')$ the random variable which corresponds to randomly choosing a prediction, and then independently selecting two protected groups according to $p_{A|\hat{Y}}$ to compare them. We now define our notion of probabilistic unfairness:

**Definition 2.2.** For $\epsilon \geq 0$ and $\delta \in [0, 1]$, we say that classifier $h$ over distribution $\mathcal{D}$ is $(\epsilon, \delta)$-probably intersectionally fair if $\Pr(U > \epsilon) \leq \delta$.

It can be seen for some given $\epsilon$ as a statement on the expected size of the population that is not being discriminated too much against. Probable intersectional fairness corresponds to searching for quantiles of $U$. We define the $\delta$-probabilistic unfairness as $\epsilon^*(\delta) = \min\{\epsilon \in \mathbb{R} \mid \Pr(U > \epsilon) \leq \delta\}$. It is the $(1 - \delta)$-quantile of $U$. We also know by definition that any classifiers over any distributions is $(u^*, 0)$-probably intersectionally fair, as we have $U \leq u^*$ with probability 1. This shows that probabilistic fairness is a relaxed version of the hard intersectional unfairness as $\lim_{\delta \to 0} \epsilon^*(\delta) = u^*$, and thus can be made arbitrarily close to intersectional fairness. In order to give more intuition on what this measure of fairness represents, we will briefly only for this paragraph consider discrimination of protected groups compared to the predictions distribution $p_{\hat{Y}}$ instead of between groups, meaning that we now measure $|\log(p_{\hat{Y}|A}/p_{\hat{Y}})|$. Suppose that a prediction model will be deployed over a population of $n$ individuals. Then if the classifier is $(\epsilon, \delta)$-probably intersectionally fair, this means that $\mathbb{E}_{A, \hat{Y}}[\sum_{i=1}^{n} \mathbb{1}[u(\hat{Y}^{(i)}, A^{(i)}) > \epsilon]]$ the expected number of people that faces a discrimination more than $\epsilon$ is less than $n\delta$. This allows us to measure and control the size of the population that may face a difference in treatment that would be deemed too high. It corresponds to the notion of fairness we were searching for. For more comparisons between these different notions, see Appendix A.3.

As a remark, looking at $\mathbb{E}[U]$, it can serve as a lower bound of $u^*$ because $\mathbb{E}[U] = \sum_{y,a,a'} p_{\hat{Y},A,A'}(y, a, a')u(y, a, a') \leq u^*$. This represents the average discrimination in a population between two protected groups. This is weaker than the notion presented above and is only mentioned in passing.

Probabilistic fairness can be especially relevant in the context where $A$ are continuous sensitive attributes. Indeed, even for a very basic multivariate normal distribution on $A$, we will end up with $u^* = \infty$ which is unhelpful. Yet by considering this notion of probabilistic fairness we end up with finite (hence comparable) measures of unfairness where the discriminated population size can be explicitly controlled; see Appendix A.4 for some examples. All in all, this notion of probabilistic unfairness, beyond its main interest of being a relaxed version of intersectional unfairness, could be in itself helpful for decision makers.

## 3 Measures of Independence and Theoretical Bounds

We now focus on providing valid $(\epsilon, \delta)$ couples for probable intersectional fairness. First note that while the intersectional unfairness $u^*$ is hard to estimate, it is much easier to estimate the marginal unfairness $u^M$. The work done by [22] in the different setting of weighted unfairness, however, shows through experiments that across multiple classifiers and data-sets, $u^*$ and $u^M$ can be uncorrelated, correlated, or even equal. Building on this observation, we would like to approach $u^*$ using marginal quantities estimable for reasonably-sized data-sets.

### 3.1 Intersectional Unfairness with Independence

Since the intersectional unfairness takes into account the interactions between all the protected attributes $A_k$, one could guess that if the $A_k$ are mutually independent, this implies that $u^*$ is close to $u^M$. Our first result is not far from this intuition, but we also need to take into account the influence from the classifier $h$. Indeed, even if the protected attributes are independent, since the classifier makes predictions based on $X$ which may encode redundant information from some $A_k$, there can be interaction between those protected attributes through the classifier. See Appendix B.1 for a counter example with the independence of the $A_k$ only but no clear relationship between marginal and intersectional fairness.

**Proposition 3.1.** *If the protected attributes $A_k$ are mutually independent and mutually independent conditionally on $\hat{Y}$, then*

$$u^* = \sup_{y \in \mathcal{Y}} \sum_{k=1}^{d} \sup_{(a_k, a'_k) \in \mathcal{A}_k^2} u_k(y, a_k, a'_k) \leq \sum_{k=1}^{d} u_k^*. \tag{3}$$

*Sketch of proof.* The main idea is to decompose $p_A$ and $p_{A|\hat{Y}}$ as their products of marginals using the independence assumptions, and using the fact that the sup taken over a product of functions with independent variables is distributed over the product. The inequality is obtained because the sup of a sum is smaller than the sum of the sup. See proof in Appendix B.2. □

This theorem gives us a first sense on how intersectional unfairness relates with marginal unfairness in some contexts. This shows us that if the independence conditions are fulfilled, then $u^*$ becomes easy to estimate. What we provide here are conditions and a equation to derive a direct relationship between the intersectional unfairness and the marginal unfairness of each $A_k$. These are unfortunately too strong conditions to actually expect and are almost never randomly satisfied, but they help us give insight into the relationship between marginal and intersectional fairness. It also drives the analysis conducted in the next sub-section. We would like to relax the independence criteria while still using marginal information from the problem.

### 3.2   Bounds on Probable Intersectional Fairness

In order to bound the probable intersectional unfairness and relate it with the strictly independent case, we want to use some measure of independence. We want to bound in probability the joint probability density $\Pr(A\!=\!a)$ with the product of its marginals $\prod_{k=1}^{d} \Pr(A_k\!=\!a_k)$. We will use one of the possible multivariable generalization of Mutual Information known as Total Correlation [29]:

$$C(A)\!=\!\mathbb{E}_A\Big[\log\Big(\frac{p_A(A)}{\prod_{k=1}^{d} p_{A_k}(A_k)}\Big)\Big]=\sum_{a\in\mathcal{A}}p_A(a)\log\Big(\frac{p_A(a)}{\prod_{k=1}^{d} p_{A_k}(a_k)}\Big)=\Big(\sum_{k=1}^{d} H(A_k)\Big)-H(A), \quad (4)$$

where $H(A)$ is the Shannon Entropy of $A$. Similarly we define the conditional total correlation as $C(A\,|\,\hat{Y}) = \mathbb{E}_{A,\hat{Y}}[\log(p_{A|\hat{Y}}(A\,|\,\hat{Y})/\prod_k p_{A_k|\hat{Y}}(A_k\,|\,\hat{Y}))] = (\sum_{k=1}^{d} H(A_k\,|\,\hat{Y})) - H(A\,|\,\hat{Y})$ where $H(A\,|\,\hat{Y})$ is the conditional entropy of $A$ given $\hat{Y}$. Note that both can also be written in terms of a KL or expectation in $\hat{Y}$ over conditional KL divergence, which means that $C(A)\!\geq\!0$ and $C(A\,|\,\hat{Y})\!\geq\!0$. From these measures of independence, we intuitively define the following two random variables, $L = \log(p_A(A)/\prod_k p_{A_k}(A_k))$ and $L_y = \log(p_{A|\hat{Y}}(A\,|\,\hat{Y})/\prod_k p_{A_k|\hat{Y}}(A_k\,|\,\hat{Y}))$. By definition we have that $\mathbb{E}[L]\!=\!C(A)$ and $\mathbb{E}[L_y]\!=\!C(A\,|\,\hat{Y})$. We denote $\sigma$ and $\sigma_y$ the standard deviation of these two variables. We have the following property:

$$\perp\!\!\!\perp_{k=1}^{d} A_k \Leftrightarrow C(A) = 0 \Leftrightarrow \sigma = 0 \quad \text{and} \quad \perp\!\!\!\perp_{k=1}^{d} A_k|\hat{Y} \Leftrightarrow C(A|\hat{Y}) = 0 \Leftrightarrow \sigma_y = 0. \quad (5)$$

The equivalence between independence and $C(A)\!=\!0$ comes from rewriting $C(A)$ as a KL and the fact that $\mathrm{KL}(P\|Q)\!=\!0$ if and only if $P\!=\!Q$ almost everywhere. For $C(A\,|\,\hat{Y})\!=\!\mathbb{E}_y[\mathrm{KL}(p_{A|\hat{Y}=y}\|\otimes p_{A_i|\hat{Y}=y})]$ we also use that the expectation of a positive random variable is 0 if and only the variable is 0 almost everywhere. When $\sigma\!=\!0$ then $L\!=\!c$ is a constant which means that $p_A\!=\!\prod_k p_{A_k} e^c$, and using that the probabilities must sum to 1 we have $e^c\!=\!1$ hence $L\!=\!c\!=\!0$. The same arguments apply for $\sigma_y$. We denote $I(V,W)\!=\!H(V) - H(V\,|\,W)$ the mutual information between a variable $V$ and $W$. With these definitions, we can now derive the following theorem which bounds the probable intersectional fairness with independence measures and functions of marginal densities:

**Theorem 3.2.** *For $\delta \in (0,1]$, any classifier $h$ over a distribution $\mathcal{D}$ is $(\epsilon_1, \delta)$ and $(\epsilon_2, \delta)$-probably intersectionally fair with*

$$\epsilon_1 = 2\sqrt{2}\frac{s^*}{\sqrt{\delta}} + \sup_{y\in\mathcal{Y}}\Big\{\sum_{k=1}^{d}\sup_{(a_k,a_k')\in\mathcal{A}_k^2} u_k(y, a_k, a_k')\Big\} \quad (6)$$

$$\epsilon_2 = \sqrt{2}\frac{s^*}{\sqrt{\delta}} + \gamma + \sup_{y\in\mathcal{Y}}\Big\{\sum_{k=1}^{d}\log\Big(\frac{p_{\hat{Y}}^{1-1/d}(y)}{\inf_{a_k\in\mathcal{A}_k} p_{\hat{Y}|A_k}(y\,|\,a_k)}\Big)\Big\} \quad (7)$$

*where* $\quad s^* = (\sigma^{2/3} + \sigma_y^{2/3})^{3/2} \quad$ *and* $\quad \gamma = C(A) - C(A\,|\,\hat{Y}) = \Big(\sum_{k=1}^{d} I(A_k,\hat{Y})\Big) - I(A,\hat{Y}).$ (8)

*Sketch of proof.* We apply Chebyshev's inequality to $L$ and $L_y$ for some introduced parameters $\alpha$ to bound the tails of these random variables, while making sure that overall the probability bounds stay larger than $1 - \delta$. We can then compute inequalities on $p_A$ and $p_{A|\hat{Y}}$, and take the inf for $a$ and sup for $\alpha$. This leads to a constrained minimization problem that can be solved, which yields $s^*$. The full proof is in Appendix B.3. For $\epsilon_2$ we additionally use that $p_{\hat{Y}|A} \leq 1$ as $\mathcal{Y}$ is discrete. □

We observe that both $\epsilon_1$ and $\epsilon_2$ are composed of one term in $s^*$ related with the $\delta$-confidence, and a quantity with marginal information. Aditionnaly $\epsilon_2$ also includes a term in $\gamma$ that corresponds to some form of mutual information correction. We can control the confidence in this bound with the parameter $\delta$. Because $s^* = 0$ if and only if $\sigma = \sigma_y = 0$ and combined with (5) we can see that $s^*$ somewhat measures how far we are from the conditions of Proposition 3.1. With $\epsilon_1$ we see that when $s^*$ goes to zero, we recover exactly the conditions of Proposition 3.1.

In order to prove Theorem 3.2, we used Chebyshev's inequality. We can derive a similar proof for other concentration inequalities, specifically with Chernoff bounds through the estimation of the moment generating function, which often leads to tighter bounds. However this leads to harder quantities to estimate in addition to having to solve a non-convex optimization problem, see Appendix B.4.

To conclude this section we provide additional intuition on the relationship between marginal and intersectional fairness. We can then derive the following corollary from the proof of the above Theorem:

**Corollary 3.3.** *Denoting* $(\Omega, \mathcal{T}, \mathrm{Pr})$ *the probability space on which* $(\hat{Y}, A, A')$ *is defined, there exists an event* $F$ *so that for* $f(y, a) = \prod_{k=1}^{d}(p_{\hat{Y}|A_k}(y \mid a_k)/p_{\hat{Y}}(y))$ *we have for* $U$ *the random unfairness:*

$$-\frac{2\sqrt{2}s^*}{\sqrt{\delta}} + \sup_{\omega \in F} \left| \log\left(\frac{f(\hat{Y}, A')}{f(\hat{Y}, A)}\right)(\omega) \right| \leq \sup_{\omega \in F} U(\omega) \leq \frac{2\sqrt{2}s^*}{\sqrt{\delta}} + \sup_{\omega \in F} \left| \log\left(\frac{f(\hat{Y}, A')}{f(\hat{Y}, A)}\right)(\omega) \right|, \quad (9)$$

*with* $\mathrm{Pr}(F) \geq 1 - \delta$.

The proof can be found in Appendix B.3. This means that there is a fraction of the relevant pairs population of size bigger than $1 - \delta$, for which we can give an interval for the maximum random unfairness over this fraction $F$. This interval is centered and reduce around a unique quantity as $s^*$ goes to 0, with $s^*$ and $\delta$ determining the length of this interval. When $\delta$ goes to 0, we have $\mathrm{Pr}(F) \to 1$ hence $\sup_{\omega \in F} U(\omega) \to u^*$ because we are dealing with finite random variables. Notice also that when both $\delta$ and $s^*$ go to 0, we recover Proposition 3.1 as $\sup_{\omega \in F} |\log(f(\hat{Y}, A')/f(\hat{Y}, A))(\omega)|$ goes toward the quantity derived in this Proposition when $\delta$ goes to 0.

### 3.3 Estimation of the measures of independence

Theorem 3.2 trades the precise estimation of $u^*$ with an upper bound, but with much easier quantities to estimate. More specifically, as they are information measures, we can leverage the extensive literature on statistical estimators and entropy estimation. We can intuitively see that the estimation of $s^*$ and $\gamma$ will be easier to handle because even the estimation with the empirical distribution $\hat{p}_{A,\hat{Y}}$ is always well defined, and is a Maximum Likelihood Estimator (MLE) as continuous functions of MLE. They are well defined because $s^*$ and $\gamma$ are functions of entropies and of the quantities $Q(P) = \sum_i p_i \log(p_i)^2$ for a probability distribution $P$, which is finite event for $p_i = 0$ because $x \mapsto x \log(x)$ and $x \mapsto x \log(x)^2$ are continuous at 0. Contrarily to $u_B$ we do not have to use any prior to obtain a well defined estimator. In addition, using the delta method on the sum of entropies, for which the MLE is asymptotically normal (See [27] 3.1), shows that $\hat{\gamma}$ is asymptotically normal. For more information on the estimation of entropy, mutual information or total correlation we defer to [27, 26, 3, 6, 15] to name but a few. Moreover even with the very simple MLE, we can obtain $L_2$ error upper-bounds for $H(P)$ in $\mathcal{O}(\log(|P|)^2/n)$ where $|P|$ is the number of outcomes for a discrete distribution $P$ [20]. This bound depends only on the number of outcomes (supposed known), and not the actual distribution. Using the same tools, we derive a rough error bound for $Q(P)$:

**Proposition 3.4.**

$$\mathbb{E}[(Q(P) - Q(\hat{P}))^2] = \mathcal{O}\left(\frac{\log^4(n)}{n}\right). \quad (10)$$

*Sketch of proof.* We apply the methods described in [20] that bounds the bias using approximation theory for Bernstein polynomials and bounds the variance using the Efron-Stein inequality. See proof in Appendix B.6. $\qquad \square$

More efficient estimators can be created using methods of [19], nevertheless the main interest of this proposition is to show that these quantities have an error rate depending on the number of samples $n$, and not the number of samples per group $N_a$, which is much better.

Beyond the practical use of these inequalities and approximations, these theorems also show one crucial idea: we can relate intersectional and marginal unfairness with the help of information on the independence of the protected attributes.

# 4 Refined approximations and inequalities

In the previous section, we have derived conditions for marginal unfairness to directly relate to $u^*$, and bounds on probable intersectional fairness. We now would like to propose an approximation of $u^*$ using similar ideas. Looking at (6), (3), and indirectly through Corollary 3.3 it seems natural to propose as one possible approximation of $u^*$ the following quantity:

$$u_I = \sup_{y \in \mathcal{Y}} \sum_{k=1}^{d} \sup_{(a_k, a_k') \in \mathcal{A}_k^2} u_k(y, a_k, a_k'). \tag{11}$$

For the rest of the article, we thus now only focus on $s^*$ and $u_I$. Compared with $u_B$ the estimator with Bayesian prior, it does not depend on a prior parameter, and is usually well defined as we only need $N_{a_i, y}$ the number of samples per $a_i$ and $y$ to be strictly positive instead of all $N_{a,y}$. However this estimator of $u^*$ is not consistent. We will show that the previous bounds can be improved and that we can make our estimator consistent by gradually grouping together the protected attributes as the number of samples increases.

## 4.1 Grouping protected Attributes together

Until now, we have always decomposed the protected attributes $A$ on their marginals $A_i$. However it may be that we have more than just marginal information available. Take the example of 4 protected attributes $A = (A_1, A_2, A_3, A_4)$. For a set $t \subseteq \{1, 2, 3, 4\}$, we define $A_t = (A_k)_{k \in t}$. We may not have enough data to compute the full intersectional unfairness, but it may be possible to compute it for the grouped protected attributes $A_{\{1,2\}} = (A_1, A_2)$ and $A_{\{3,4\}}$. We can use the same decomposition as we did before on the new marginals attributes (which corresponds to flattening $A_1$ and $A_2$ together) with support $\mathcal{A}_{\{1,2\}} = \mathcal{A}_1 \times \mathcal{A}_2$ and $\mathcal{A}_{\{3,4\}} = \mathcal{A}_3 \times \mathcal{A}_4$.

More generally, let $q$ be a partition of $\{1, ..., d\} = [d]$. For a partition $q$, we denote $A^{(q)} = (A_t)_{t \in q}$. This is only a different way to group together the marginal attributes, and is the same as $A$. Whenever quantities are changed according to some partition $q$, it will be indicated with $(q)$. For each of the new marginal attributes defined by a set $t$ of $q$, the new marginal unfairness $u_t^*$ corresponds to the intersectional unfairness of the $(A_k)_{k \in t}$. If the $A_t$ are independent, and independent conditionally on $\hat{Y}$, we can apply Proposition 3.1 and obtain directly $u^*$ through the newly defined marginals. If we relax the independence conditions, the same arguments of the previous section still apply, and we can look at the bounds and approximations defined by these new marginal densities. We denote the new approximation with partition $q$ by $u_I^{(q)} = \sup_{y \in \mathcal{Y}} \sum_{t \in q} \sup_{(a_t, a_t') \in \mathcal{A}_t^2} u_t(y, a_t, a_t')$ where we are using the new marginals defined by $q$. If we use the partition $q$ of singletons then $u_I^{(q)} = u_I$, and if we use the trivial partition (the whole set) then $u_I^{(q)} = u^*$. The constraints of independence for these new marginals should be more feasible than the original marginals, hence it is possible that the $A_t$ fulfill the independence conditions, even if the $A_k$ do not (the trivial partition is such an example). If we have enough data to compute the marginal densities derived from $q$ and the $A_t$ fulfill the independence conditions, we can then compute $u^*$ through the partition $q$. Of course most of the time the independence conditions are not satisfied satisfied for a partition $q$. Nonetheless because $s^*$ measures how far we are from the independence conditions, we can more carefully select a partition among those for which we can compute the new marginal densities.

## 4.2 Efficient Partition Selection

Let $\mathcal{Q}$ be the set of all *feasible* partitions $q$, that is $\mathcal{Q} = \{q \in \mathcal{P}([d]) \mid \forall t \in q, \forall (a_t, y) \in \mathcal{A}_t \times \mathcal{Y}, N_{a_t, y} > 0\}$ with $\mathcal{P}([d])$ the set of all partitions of $[d]$. This set represents the set of partitions for which we can compute the newly defined marginals without having to use a prior parameter. Note that $\mathcal{Q}$ is a random set that converges to $\mathcal{P}([d])$ almost surely as the number of samples $n$ increases. If $q' \in \mathcal{Q}$, then any partitions $q$ finer than $q'$ (meaning that any element of $q$ is a subset of an element of $q'$) is in $\mathcal{Q}$ as well. We will say that $q$ can be merged further if there exists a partition $q' \in \mathcal{Q}$ so that $q$ is finer than $q'$. Note that the choice of a partition $q$ does not change the value of $u^*$ but only that of $u_I^{(q)}$. We therefore want to find a good feasible partition $q$ in $\mathcal{Q}$ so that we can expect

heuristically $|u_I^{(q)} - u^*|$ to be the lowest among the partitions. There are two criteria that should help us decide which partition $q \in \mathcal{Q}$ to choose from.

**Algorithm 1** Greedy Partition Finder

> **input:** Protected attributes data and occurrences of $\hat{Y}$
> **require:** The partition of singletons is feasible
> $q^* \leftarrow$ the partition of singletons
> **repeat**
> > $\mathcal{M} = \{\{t_1 \cup t_2\} \cup q^* \setminus (\{t_1\} \cup \{t_2\}), (t_1, t_2) \in q^{*2}, t_1 \neq t_2\}$
> > $s_{\min}^* \leftarrow +\infty$
> > **for** $q$ in $\mathcal{M}$ **do**
> > > **if** $q$ is feasible and $s^*(q) < s_{\min}^*$ **then**
> > > > $(s_{\min}^*, q^*) \leftarrow (s^*(q), q)$
> > >
> > > **end if**
> >
> > **end for**
>
> **until** $\mathcal{M} = \emptyset$ or $s_{\min}^* = \infty$ (Nothing possible to merge)
> **return:** $q^*$

If a partition $q'$ is coarser than $q$ (which means that $q$ is finer than $q'$), then reasonably the approximation is better with $q'$ than $q$. The reasoning is that by taking coarser partitions, we are taking more interactions between the protected attributes into account. For example the coarsest partition which is the whole set gives us the intersectional unfairness as mentioned earlier. However because the 'finer-than' relationship is only a partial order, we are not able to choose between any two sets. Because Theorem 3.2 seems to hint that there is a relationship between the error $|u_I^{(q)} - u^*|$, and the distance to the independence conditions $s^*$, the second criterion will be to select the partitions $q$ with the smallest $s^*(q)$ defined as $s^*$ but taking the marginals in $q$. These two criteria are closely linked. Selecting coarser partitions does tend to yield partitions with smaller $s^*$, but not always. We give some details on relationship between $s^*(q)$ of a partition $q$ compared to a coarser one in Appendix C.2. More crucially, finding a good partition with a small $s^*(q)$ will also improve our inequalities as they are a function of $s^*(q)$ which decreases on average as shown in Figure 5 as the number of sample grows (and as the partitions get coarser).

In principle, finding the best partition according to our criteria requires enumerating all feasible partitions which is computationally intractable. Instead we propose a greedy heuristic that we describe in Algorithm 1. We start from the finest partition (the partition of singletons), look at all the feasible partitions (with enough data) that can be obtained from merging two elements of the current partition, select the one with the smallest $s^*$, and repeat until there are no coarser partitions with enough data. Note that when we want to verify that there is enough data available, we may need to do it multiple times for the same subset of protected attributes. This is an expensive call so it is more efficient to do memoization and remember if there is enough data available for a given subset once encountered, which we do using a hash table to reduce the lookup time. We denote this partition $q^*$. We have the following property with the proof in Appendix C.1:

**Proposition 4.1.** *The estimator $u_I^{(q^*)}$ is a consistent estimator of $u^*$.*

This proposition shows that $u_I^{(q^*)}$ is relevant in estimating $u^*$, while not needing to use a Bayesian prior with parameters that may overwhelmingly affect the estimation. Note that instead of using $\mathcal{Q}$ which ensures that $N_{a_t,y} > 0$, we can instead use $\mathcal{Q}_\tau$ for $\tau \in \mathbb{N}$ which ensures that for any $q \in \mathcal{Q}_\tau$, $N_{a_t,y} > \tau$ for $t \in q$.

## 5 Experiments

In this section, we present experimental results that show how our inequalities and approximations perform on real and synthetic data-sets, and compare their estimation error rates as the number of samples grow. All the code used in our experiments can be found in the supplementary material or at https://github.com/mathieu-molina/BoundApproxInterMargFairness.

### 5.1 Data-sets and processing

In order to compare how well $u_B$ and $u_I$ perform as estimators on data on datasets with a high number of protected attributes, we need to compute $u^*$ which is as discussed above inherently difficult. We will always measure the unfairness with respects to the empirical distribution of the dataset. For this empirical distribution to yield a well defined fairness measure, we need that $N_{a,y} > 0$ for all $a$ and $y$. This means that if we want to take into account a high number of sensitive attributes, we have to pick a very large dataset.

We used US Census data from 1990 [11] which contains $n = 2,458,285$ samples, and for which we identified many potential protected attributes. We then train a Random Forest binary classifier on a poverty binary label, where we weight the labels differently so as to obtain about the same number of predictions for each outcome. However we still do not have $N_{a,y} > 0$ on the whole dataset. To alleviate this issue, we will consider subsets of the protected attributes for which this is true, and we will measure fairness with respect to these subsets. We obtain about 100 different subsets with $d = 8$ protected attributes, that we denote as $D_i$ which is the original dataset where we only kept the $i$-th subset of protected attributes and the predictions $\hat{Y}$. Each of these subsets yield different values of $u^*$ and $s^*$. We pick 12 (for computational reasons) different $D_i$ with various values of $u^*$ and $s^*$. Some examples of the final protected attributes include sex, not speaking English at home, being overweight, being Hispanic, and others. We will always take $\delta = 0.1$ when relevant.

We also conduct experiments on synthetic data. We generate $(A, \hat{Y})$ probability distributions from a Dirichlet distribution, thus we can directly compute $u^*$ without dealing with a very large dataset. We take $d = 10$. This synthetic data is one of the worst case for the approximation of $u^*$ with $u_I$, as the marginal distributions are a sum of $2^{d-1}$ i.i.d. random variables that all converges to $1/2$ as $d$ grows. We therefore will not plot $u_I$ for the synthetic data (it is close to 0). Nonetheless, this synthetic data remains useful in order to compare the error rates between $u_B$ and $\hat{s}^*$ and their respective true value. We denote by $P_i$ a generated probability distribution. We generate 12 of them.

## 5.2   Experiments Results

We first want to compare the convergence rate of $u_B$, $s^*$ and $u_I$ to their asymptotic value. To do that, and because they can take different values, we compute for each estimator $\hat{T}_n$ that converges in probability to $T$ the relative expected $L_2$ error rate $L_2^r(\hat{T}_n) = \mathbb{E}[(\hat{T}_n - T)^2/T^2]$. We fix a number of available samples $n$ from 100 to $2,000$, and we sample without replacement from the datasets. From these available samples, we compute all our estimators. We denote by $\hat{u}_I$ the estimator of $u_I$ computed with the empirical marginal densities for $n$ samples. In order to compute $L_2^r$, for each subset and each sample size $n$, we sample from $D_i$ and $P_i$ 20 times for each fixed number of samples $n$.

We see in Figure 1 on the left-most plot, that $\hat{u}_I$ is reasonably close to $u^*$ on average. Still the gap between $\epsilon^*$ and $\epsilon_1$ is quite big. Other bounds such as $\epsilon_2$ or with other concentration inequalities are generally a bit more efficient but still loose, nevertheless we focus here on comparing the error rates between $s^*$ and $u_B$, and on how $u_I$ performs. The other two plots look at the $L_2^r$ for the various estimators, with the middle one being with the real datasets $D_i$, and the right on the synthetic datasets $P_i$. We can see that $\hat{s}^*$ and $\hat{u}_I$ converges much faster than $u_B$. Moreover, it seems that the difference in error rate will only grow bigger as $d$ increases, as there is a bigger gap for $P_i$. We can also see that $u_B$ is unreliable, because the error rate varies a lot depending on $\alpha$, and can even increase. This is because the parameter $\alpha$ dominates the computation of $u_B$ as discussed earlier.

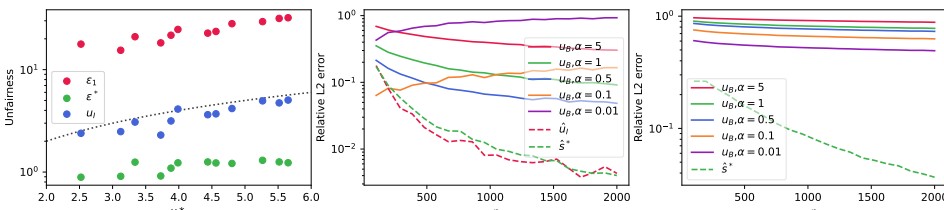

Figure 1: On the left-most plot, each point represents one real dataset $D_i$, and we compare $\epsilon_1$, $\epsilon^*$, and $u_I$ with $u^*$. The dotted line corresponds to the equation $x = y$ for reference. The middle plot describes the average over the $D_i$ of $L_2^r$ as $n$ increases for $\hat{u}_I$, $\hat{s}^*$, and $\tilde{u}^*$. $u_B$ is computed for multiple values of $\alpha$. The right-most plot is similar, but uses the synthetic datasets $P_i$.

We now conduct similar experiments, but this time using partitions. We can see in the middle plot of Figure 2 that $\hat{u}_I^{(q^*)}$ performs better. The choice of $\tau$ the count threshold for grouping always gives reasonable approximations, with $\tau = 1$ being close to $u_B$, and $\tau$ big makes it close to $u_I$. Most importantly, the apparent good error rate of $u_B$ is merely an artifact of the current range of $u^*$ being above the starting values of $u_B$ for these $\alpha$. It is clear that $u_B$ is unreliable by looking at the left plot in Figure 2: the estimation with $u_B$ at $n = 2000$ for different values of $\alpha$ varies very little when

$u^*$ varies (it is almost not a function of $u^*$). This means that $u_B$ depends very little on the data for low amount of samples. Even if it is not perfect, $u_I^{(q^*)}$ still has better performance and is more coherent. We note that the approximation performs well comparatively only when $d$ is high, and considering more sensitive attributes should make an even bigger difference. These results combined with Proposition 4.1 show that $u_I^{(q^*)}$ is a relevant estimator of $u^*$ with scarce data and high number of protected attributes. Concerning $s^*(q^*)$ the right-most plot shows that while it is not completely monotone, $s^*(q^*)$ does decreases on average when using partitions as the number of sample increases. The upper bound will become tighter as $n$ grows, which will make bigger groupings of protected attributes possible.

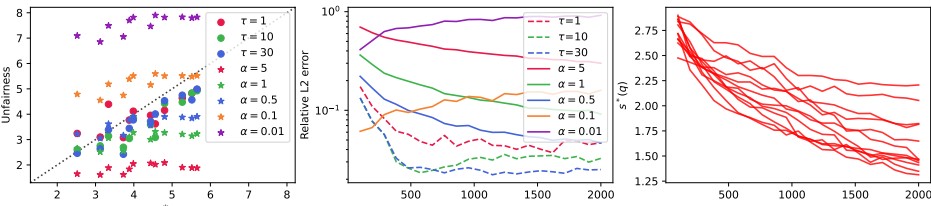

Figure 2: Each point of the same color and shape represent the estimation for one dataset $D_i$. The estimations are computed for $n = 2000$. The middle plot is the same as above, but with $\hat{u}_I^{(q^*)}$ this time. The rightmost plot is the average evolution of $s^*(q^*)$ for $\tau = 10$ as $n$ increases.

## 6   Discussion

In this work, we presented new methods to approximate and to bound (in high probability) a strong intersectional unfairness measure, based on statistical information computable from a reasonable dataset. Our results highlight the key role of independence of the protected attributes conditionally to the classifier, and propose to approach it via a smart grouping of some attributes—which our theoretical bound allows us to compute via an efficient heuristic.

Our experiments show that the approximations proposed here perform reasonably well for data-sets with a high number of protected attributes, but that our bounds are not very effective. However their main interest is that it gives insight into the link between marginal and intersectional fairness, which was the main goal of this work. It also helps us derive the proposed approximation. We expect that more effective bounds could be derived for our notion of probabilistic fairness, for instance by making additional assumptions on the distribution, but presumably without an explicit dependence on independence measures and marginal densities, making the link between marginal and intersectional fairness harder to see.

In order to train fair models using the proposed approximations or bounds of this paper, we can use soft counts to compute the empirical densities (based on the classifier score for instance) as suggested in [14]. This makes the approximations and bounds differentiable, and ensure that we can apply gradient based methods so as to solve a constrained or penalized optimization problem using these quantities.

We hope that our approach will enable the development of improved bounds, raise interest in the proposed notion of probabilistic unfairness which we think is crucial to the development of fair algorithms, as well as the use of our approximations to penalize classifiers in order to train intersectionally fair classifiers.

### Acknowledgments

This work has been partially supported by MIAI @ Grenoble Alpes (ANR-19- P3IA-0003), by the French National Research Agency (ANR) through grant ANR-20-CE23-0007, and by TAILOR (a project funded by EU Horizon 2020 research and innovation programme under GA No 952215). The authors are hosted at the CREST lab (CNRS, GENES, Ecole Polytechnique, Institut Polytechnique de Paris).

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
