# OpenReview forum: "Bounding and Approximating Intersectional Fairness through Marginal Fairness"
_NeurIPS.cc/2022/Conference — NeurIPS 2022 Accept_

### Official Review · Reviewer_U9ky · 2022-07-02

**Rating:** 6
**Confidence:** 3
**Soundness:** 3 good
**Presentation:** 2 fair
**Contribution:** 3 good

**Summary:**

This paper studies how to estimate intersectional fairness when there are a large number of protected attribute combinations. The paper first show that if the protected attributes are mutually independent and mutually independent conditionally on the prediction, the intersectional fairness can be easily computed from the marginal fairness. Then, the paper relaxes the independent assumptions and provides bounds on probable intersectional fairness. After that, the paper offers a bound directly on the intersectional fairness. The idea is to find a partition in order to get closer to the independence conditions so that the intersectional fairness computed under the independence assumptions could be a good approximation. Experiments using real-world and synthetic data show that the proposed bounds are more reliable than the Bayesian estimator.

**Questions:**

1.	Are u^q_ind upper bounds or lower bounds? Or both cases are possible? In that case, why is u_ind in Figure 1 constantly smaller than u*?

2.	Are epsion_1 and epsilon’_1 upper bounds? What is epsilon* in Figure 1? Any explanation of why epsilon* is smaller than u*?

3.	The authors said that the proposed approach will enable the use of the approximations to penalize classifiers in order to train intersectionally fair classifiers. Is that possible to include some discussions about the differentiability of the proposed method that is necessary for the approximations to be integrated into gradient-based optimization?


**Limitations:**

Yes.

**Strengths And Weaknesses:**

Strength

This paper systematically studies the bounding and approximating of intersectional fairness through marginal fairness. It provides a sufficient condition for accurately computing intersectional fairness from marginal fairness. Then, it provides both probabilistic bounds and hard approximations of the intersectional fairness, which are easily computable from the data.

Weakness

The paper does not provide a convincing example showing why ensuring intersectional fairness for a very large number of subgroups (e.g., 100 subgroups) is a meaningful and practical requirement for decision-makers. Although this paper studies the measurement of fairness not the achievement of fairness, the measurement is meaningful only when we have the requirement to achieve fairness.

There are too many notations, which make the paper difficult to follow.

---

> ### Author Response · Authors · 2022-08-02
> **Response to the reviewer**
>
> Dear reviewer, we thank you for your careful read of our submission and for this review.
>
> 1) To give an example of why wanting to ensure fairness for many attributes may be desirable, French law (https://www.legifrance.gouv.fr/codes/article_lc/LEGIARTI000033461473/) identifies 25 protected attributes (pregnancy, wealth, religion, sex, handicap, health to name but a few) for which discrimination for access to health, education, employment and others critical areas is forbidden. While we currently do not see many applications using many protected groups, it might be because of the inherent difficulty of assessing fairness in this context. Note that even having 6 or 7 binary sensitive attributes makes it quite challenging to estimate fairness for moderately sized data-sets.
>
> 2) The quantities $u^q_{ind}$ or u_ind by themselves are neither an upper or lower bound of u* (but $\epsilon_1$ which contains $u_{ind}$ is an upper bound of $\epsilon^*$). As can be seen in the leftmost plot of Figure 2, there are values of $u_{ind}^q$ above and below the diagonal line which represents u*. The fact that for $u_{ind}$ most of the points are below u* is coincidental. There is nonetheless a relationship between these quantities: from the proof of Theorem 3.2 we can directly derive the following additional property (using the inequalities already derived for $\Pr(\hat{Y} \mid A)$:  there is a fraction F of size at least $1-\delta$ of the population so that the intersectional unfairness when considering only this fraction of the population $u^*_F$ is in the interval
>
> $[u_{ind}-2\sqrt{2}s^*/\delta, u_{ind}+2\sqrt{2}s^*/\delta]$. Hence, for a fixed \delta we see that as s* diminishes, the interval reduces around $u_{ind}$.
>
> 3) The quantity $\epsilon^*$ is defined as the smallest valid value of $\epsilon$ for a fixed $\delta$, and can be seen as the quantile of random variable $U$ defined as the function u taking as input the random variables $A$ and $\hat{Y}$. Therefore as a quantile of the distribution of $U$, it is always smaller than the sup of the support of the distribution which is u*. Both $\epsilon_1$ and $\epsilon’_1$ are upper bounds of $\epsilon^*$, as $\epsilon^*$ is the smallest possible value of $\epsilon$ for $\delta$ fixed.
>
> 4) In order to make the quantity $u_{ind}$ differentiable so as to use gradient based method for optimization, we can use soft counts (based on the score of the classifier for example) instead of the actual prediction to compute the densities as suggested in Foulds et al. 2020 [14]. We will include this comment.
>
> 5) We will simplify the notations accordingly to make the paper easier to follow.
>
> We thank the reviewer for giving us the opportunity to clarify some of the motivations as well as the relationship between some of these quantities.

---

> > ### Comment · Reviewer_U9ky · 2022-08-09
> > **Comment**
> >
> > I would like to thank the authors for their responses which address most of my concerns.

---

### Official Review · Reviewer_WyGu · 2022-07-10

**Rating:** 5
**Confidence:** 4
**Soundness:** 2 fair
**Presentation:** 2 fair
**Contribution:** 2 fair

**Summary:**

This paper presents new methods to approximate and bound intersectional unfairness. Intersectional fairness is connected with marginal fairness with the mutual independence and conditional independence of protected attributes given predictions. Then the dependence conditions are relaxed via the quantification of total correlations and their variance. A grouping technique is provided to relax the mutual independence of A to the dependence of groups of A. The experiments compare the approximation gaps and show the effectiveness of proposed approximations.

**Questions:**

NA

**Limitations:**

- The motivation is very clear. But there are too many uncertainties between u* and u_{ind}. The major concern is that the relationship between u* and u_{ind} is unclear to me in general.

- The work would be complete if it shows how the approximation is used for intersectional unfairness elimination.

**Strengths And Weaknesses:**

- The motivation is simple and clear.
- The proposed approximation connects intersectional farness and marginal fairness through independence. The new approximation and relaxed version are easy to calculate with known techniques.
- The grouping technique shows another relaxation direction.

---
- The mutual independence and conditional independence imply no more than one protected attribute is correlated with Y. Let one start with A={A_1, A_2, ... A_d} and let \hat{Y} = f(A_1, A_2, ... A_d, X). The graphical representation is A1, A2, ... A_d, X -> Y. Given A are mutually independent and the definition of d-separation, the protected attributes cannot be conditionally independent. In another word, mutual independence, conditional independence, and \hat{Y} = f(A_1, A_2,... A_d, X) cannot be achieved simultaneously. Theorem 3.1 & 3.2, as well as u_{ind}, are based on this independence assumption. It is unclear whether they are affected by the prediction function \hat{Y} = f(A_1, A_2,... A_d, X).  A straightforward impact I see is s^* cannot be zero unless no more than one protected attribute is used for prediction.

- Thm 3.2 connects u* with an upper bound containing u_{ind}. There is an intuitive relationship between u* and u_{ind}, but it is unclear why it seems **natural** to propose u_{ind} as a good approximation in theory. In another word, the relationship between u* and u_{ind} is unclear in general settings.

- Sec 4.2 and Algorithm 1: it is unclear why small s^* leads to small |u^q_{ind} - u^*|. Given that \epsilon_1 is an upper bound of u*, the relationships between u_{ind}, s^* and u* are still unclear.

- The notations are really difficult to follow. I suggest the authors revise the notations and fix typos, e.g. u_k(y, a, a') in Eq (6), A_i in Eq (25).

---

> ### Author Response · Authors · 2022-08-02
> **Response to the reviewer**
>
> Dear reviewer, we thank you for your careful read of our submission and for this review.
>
> 1) It is not correct that given this graphical model, there could be no distributions which would achieve the condition of Theorem 3.1. The proof given in the review assumes that the distribution is faithful to the graph, while there could be unfaithful distributions. Notably, if we have independent protected attributes $A_k$ and we want to obtain a completely fair model meaning that the $A_k$ are independent of $\\hat{Y}$, this would imply the conditions of Theorem 3.1. Still as pointed out s* will indeed most likely not be 0 “by chance” as there are very few unfaithful distributions. We can however likely reduce the value of s* by taking larger groupings of protected attributes. Finally, we want to stress that Theorem 3.1's main purpose is to provide an interpretation of the quantity s* and $u_{ind}$ in Theorem 3.2, and that Theorem 3.2 does not use this independence assumption and is true in general.
>
> 2) Regarding the relationship between $u_{ind}$ and u, from the proof of Theorem 3.2 we can directly derive (using the inequalities already derived for $\\Pr(\\hat{Y} \\mid  A)$) the following statement: there is a fraction F of size at least $1-\\delta$ of the population so that the intersectional unfairness when considering only this fraction of the population $u^*_F$ is in the interval
>
> $[u_{ind}-2\\sqrt{2}s^*/\\delta, u_{ind}+2\\sqrt{2}s^*/\\delta]$. Hence, for a fixed $\delta$ we see that as s* diminishes, the interval reduces around $u_{ind}$. As it is the center of this interval, this justifies taking $u_{ind}$ to provide a possible approximation to u*. We can always consider larger groupings of sensitive attributes to likely obtain smaller s which in a way interpolates between directly estimating u* and computing $u_{ind}$. We will add this discussion below Theorem 3.2, and we thank the reviewer for this remark.
>
> 3) We do not claim that a smaller s* yields a smaller $|u^q_{ind} – u^*|$ and it is indeed not true in general. However taking into account the above remark regarding the relationship between $u_{ind}$ and u*, if we have a smaller s* there exists a fraction of the population so that the maximum possible distance according to Theorem 3.2 between u^q_{ind} and u* is indeed smaller. We will change it to make it clearer and to relate it with the above comment.
>
> 4) Concerning intersectional unfairness elimination using $u_{ind}$, we do agree that this is the logical next step, but in order to keep things focused and not too long we preferred not to develop this part. Still, if we want to apply gradient based methods to solve the constrained or penalized optimization problem with $u_{ind}$, we can make u_ind differentiable by using soft counts (based on the score of the classifier for example) instead of the actual prediction to compute the densities as suggested in Foulds et al. 2020 [14]. We will include this comment.
>
> 5) We will take into account the comments about the notations and simplify them accordingly to make it easier to follow (as well as correct the typos).
>
> We thank the reviewer for correctly noting that the relationship between $u_{ind}$ and $u^*$ was not sufficiently explained and enabling us to give further details.

---

### Official Review · Reviewer_xViZ · 2022-07-12

**Rating:** 6
**Confidence:** 4
**Soundness:** 4 excellent
**Presentation:** 3 good
**Contribution:** 3 good

**Summary:**

The paper aims at the task of bounding and approximating intersectional fairness, with a main motivation of cases where multiple protected attributes exist, and a finite training from which such approximation of the statistical fairness criterion could be difficult due to scarcity of datapoints for marginal subsets which are the intersections of such variable values. The paper establishes theoretical bounds, connecting marginal fairness (for varying, single, protected attributes) and intersectional fairness, by using a measure of total correlation. Namely, the idea is to bound how far the product of the marginal measures is from being independent (across the different protected attributes), and use the bound to upper bound the intersectional measure. The paper establishes these bounds as a function of the mutual information between the made predictions and the protected attributes, and as a function of the standard deviation of the total correlation measures. The bounds serve as an inspiration then for a proposed heuristic approach for estimating the intersectional fairness, which is based on greedy selection of groupings of protected attributes, which minimize the distance from independence using the bounds, allowing for better estimation. The paper ends with an evaluation of the proposed scheme on US census data from 1990, as well as on synthetic data.

**Questions:**

1. In Theorem 3.2, for the bound on \eps_1^', inside the log parantheses, what are the variables for which the probabilities are taken?

2. I wonder, how effective are the established upper bounds in terms of bounding the intersectional fairness from a finite sample? There is no illustration or provided examples which could be helpful in assessing the significance of such bounds in practice. Should we only expect the upper bounds to be meaningful when protected attributes are very close to being independent?

**Limitations:**

1. There is no discussion of the limitations of the work in the main body of the paper. I would expect to see a more elaborate one in the summary.

2. Effectiveness of theoretical bounds - please see in the "questions" section.

**Strengths And Weaknesses:**

Originality:
Strengths:
1. The theoretical bounds, connecting intersectional fairness to marginal fairness, decomposing the problem in terms of the dependence between the protected attribute are interesting, and give conceptual insight, as well as estimation (upper) bounds that rely only on the total number of samples (not the marginal).
2. The proposed attribute grouping-based heuristic compares favorably with prior work (using a Dirichlet prior).

Quality:
The results in the paper seem to be well supported, though I have only glanced at the appendix. The relevant related work is mentioned, and code is attached in the supplementary file.

Clarity:
The paper is mostly well-written. The experiments section, and the description of the results were a bit difficult to interpret and could be improved.

Significance:
Strengths:
The problem in question is highly motivated. The ability to give reliable statistical estimation of intersectional fairness with finite data is important and could have practical implications. The proposed greedy heuristic based algorithm of grouping seems to compare favorably with formerly proposed methods.

Weaknesses:
It is, however, still not completely clear to me how useful are the established upper bounds in practical scenarios. Please see "questions" section of my review.

---

> ### Author Response · Authors · 2022-08-02
> **Response to the reviewer**
>
> Dear reviewer, we thank you for your careful read of our submission and for this review.
>
> 1) In the equation of $\epsilon_1’$ we did not explicit the variables for compactness. The $\inf$ is to be taken over $a_k \in \mathcal{A}_k$ for the function $a_k \mapsto \Pr(\hat{Y}=y \mid A_k=a_k)$ for all k, and the $\sup$ is to be taken over $y \in \mathcal{Y}$ for the whole sum with inside the log as the numerator the function $y \mapsto \Pr(\hat{Y}=y)$. We will modify this formula to make explicit the dependence on these variables.
>
> 2) Regarding the bounds effectiveness, we acknowledge that in the more general setting these upper bounds are indeed effective mainly when s* is small (note, however, that we can somewhat control that by using bigger groupings of protected attributes as seen in Appendix D Figure 7). One of the benefits of these bounds is that the connection is clearer when compared to the case with exact independence conditions, which enables us to give some intuition on the link between marginal and intersectional unfairness, as well as motivates the choice of u_ind as a reasonable approximation (see also our response to Point 2) of the next reviewer, Reviewer WyGu). We also derived in Appendix B.4 other bounds using Chernoff Bounds, which are empirically tighter as seen also in Figure 7 of Appendix D. Deriving closed form of the optimization problem, however, is more difficult, as well as guaranteeing meaningful estimation error bounds. We expect that more effective bounds could be derived, for instance by making additional assumptions on the distribution, but presumably without an explicit dependence on $u_{ind}$ which makes the link with marginal unfairness more difficult to see. As this is what we explicitly wanted to look at in this paper, we did not think that this was an important direction to pursue here.
>
> 3) Concerning the lack of discussion around the limitations of this work,  we will add the above comment as a limitation to the paper.
>
> We thank the reviewer for highlighting these missing comments.

---

> > ### Comment · Reviewer_xViZ · 2022-08-08
> > **Thank you for the response**
> >
> > I wish to thank the authors for their response. Following the response to my questions and clarifications, I tend to keep my original score.

---

### Comment · Area_Chair_4okD · 2022-08-08
**Last day of discussion**

Thanks, all (authors and reviewers!) for participating in this process.  Reviewers U9ky and xViZ, any additional clarification points or comments for the authors before we're in private chat mode?  -- AC

---

### Meta-Review · Area_Chair_4okD · 2022-08-30

**Recommendation:** Accept
**Confidence:** Less certain

**Metareview:**

This paper focuses on intersectional fairness in supervised learning, specifically on understanding the relationship between ensuring fairness on one particular sensitive attribute and on the intersection of those attributes.  A largely analytical paper, bounds are given relating the latter to exact guarantees on the former.  Reviewers took minor issue with the experimental validation and with the applicability of these bounds; I agree with the former, but do not agree with the latter.  This work adds a nice building block to the foundations of fair ML research.

**Award:**

No

---

### Decision · Program_Chairs · 2022-09-14

Accept